# Phylogeography and transmission of *M. tuberculosis* in Moldova: A prospective genomic analysis

Chongguang Yang[1,2‡], Benjamin Sobkowiak[3‡], Vijay Naidu[3], Alexandru Codreanu[4], Nelly Ciobanu[4], Kenneth S. Gunasekera[2], Melanie H. Chitwood[2], Sofia Alexandru[4], Stela Bivol[5], Marcus Russi[2], Joshua Havumaki[2], Patrick Cudahy[2], Heather Fosburgh[2], Christopher J. Allender[6], Heather Centner[6], David M. Engelthaler[6], Nicolas A. Menzies[7], Joshua L. Warren[8], Valeriu Crudu[4]*, Caroline Colijn[3‡], Ted Cohen[2‡]*

1 School of Public Health (Shenzhen), Shenzhen Campus of Sun Yat-sen University, Shenzhen, China, 2 Department of Epidemiology of Microbial Diseases, Yale School of Public Health, New Haven, Connecticut, United States of America, 3 Department of Mathematics, Simon Fraser University, Burnaby, Canada, 4 Phthisiopneumology Institute, Chisinau, Republic of Moldova, 5 Center for Health Policies and Studies, Chisinau, Republic of Moldova, 6 Translational Genomics Research Institute, Flagstaff, Arizona, United States of America, 7 Department of Global Health and Population, and Center for Health Decision Science, Harvard TH Chan School of Public Health, Boston, Massachusetts, United States of America, 8 Department of Biostatistics, Yale School of Public Health, New Haven, Connecticut, United States of America

‡ CY and BS share first authorship on this work. CC and TC are joint senior authors on this work.
* valeriu.crudu@gmail.com (VC); theodore.cohen@yale.edu (TC)

## Abstract

### Background

The incidence of multidrug-resistant tuberculosis (MDR-TB) remains critically high in countries of the former Soviet Union, where >20% of new cases and >50% of previously treated cases have resistance to rifampin and isoniazid. Transmission of resistant strains, as opposed to resistance selected through inadequate treatment of drug-susceptible tuberculosis (TB), is the main driver of incident MDR-TB in these countries.

### Methods and findings

We conducted a prospective, genomic analysis of all culture-positive TB cases diagnosed in 2018 and 2019 in the Republic of Moldova. We used phylogenetic methods to identify putative transmission clusters; spatial and demographic data were analyzed to further describe local transmission of *Mycobacterium tuberculosis*. Of 2,236 participants, 779 (36%) had MDR-TB, of whom 386 (50%) had never been treated previously for TB. Moreover, 92% of multidrug-resistant *M. tuberculosis* strains belonged to putative transmission clusters. Phylogenetic reconstruction identified 3 large clades that were comprised nearly uniformly of MDR-TB: 2 of these clades were of Beijing lineage, and 1 of Ural lineage, and each had additional distinct clade-specific second-line drug resistance mutations and geographic distributions. Spatial and temporal proximity between pairs of cases within a cluster was associated with greater genomic similarity. Our study lasted for only 2 years, a relatively short

**Data Availability Statement:** The genomic data have been made available through GenBank (PRJNA736718, https://www.ncbi.nlm.nih.gov/bioproject/PRJNA736718). Additional data used in

the analysis (with the exception of location data which cannot be provided because of the small number of participants at locations would allow linkage to individual participants), are provided as a csv in the Supporting information.

**Funding:** This study was made possible by the generous support of the American people through the United States Agency for International Development (USAID) through the TREAT TB Cooperative Agreement No. GHN-A-00-08-00004 (TC, CC, and VC). CY received funding from the Nation Institutes of Health- Clinical and Translational Science Awards (CTSA) program No. UL1 TR001863. The funders had no role in study design, data collection and analysis, decision to publish, or preparation of the manuscript.

**Competing interests:** The authors have declared that no competing interests exist.

**Abbreviations:** CTAB, cetyl trimethyl ammonium bromide; HPDI, highest posterior density interval; IQR, interquartile range; MDR, multidrug resistance; MDR-TB, multidrug-resistant tuberculosis; M–L, maximum–likelihood; NGS, next generation sequencing; OR, odds ratio; RR, risk ratio; RRDR, rifampicin resistance determination region; SNP, single nucleotide polymorphism; TB, tuberculosis; TMRCA, time to the most recent common ancestor; XDR, extensive drug resistance.

duration compared with the natural history of TB, and, thus, the ability to infer the full extent of transmission is limited.

## Conclusions

The MDR-TB epidemic in Moldova is associated with the local transmission of multiple *M. tuberculosis* strains, including distinct clades of highly drug-resistant *M. tuberculosis* with varying geographic distributions and drug resistance profiles. This study demonstrates the role of comprehensive genomic surveillance for understanding the transmission of *M. tuberculosis* and highlights the urgency of interventions to interrupt transmission of highly drug-resistant *M. tuberculosis*.

## Author summary

### Why was this study done?

- The transmission of multidrug-resistant tuberculosis (MDR-TB) poses a major challenge for tuberculosis (TB) control in several countries, but a detailed understanding of the local dynamics of TB and MDR-TB transmission in these high MDR-TB burden settings has been elusive.

- The increasing availability of whole genome sequencing, and the development of new statistical approaches for combining spatial, epidemiological, and genomic data to infer transmission, offers new opportunities to identify TB transmission with high resolution.

### What did the researchers do and find?

- We prospectively enrolled all individuals with incident culture-positive TB from the Republic of Moldova, a high MDR-TB burden setting, between January 2018 and December 2019 and sequenced a diagnostic *Mycobacterium tuberculosis* isolate from each individual.

- We found that that nearly all extant MDR-TB in Moldova is likely the result of recent transmission and that multidrug resistance (MDR) is highly concentrated within 2 *M. tuberculosis* lineages (Beijing and Ural).

- Phylogeographic analyses revealed geographically distinct patterns of transmission for the Beijing MDR strains, which were predominantly localized within the Transnistrian region to the east of the country, while Ural MDR strains were less geographically restricted.

- Each putative MDR-TB transmission cluster had distinct second-line drugs resistance-conferring mutations. Population genetic analyses revealed both long periods of local population expansion as well as more recent introduction of specific MDR-TB strains into the country.

**What do these findings mean?**

- To our knowledge, this is first study to comprehensively sequence all *M. tuberculosis* isolates from an entire high MDR incidence country and offers unique insights into the complexity MDR-TB transmission in Moldova.

- Local transmission of distinct highly drug-resistant *M. tuberculosis* strains suggests that public health and clinical interventions tailored to address such local heterogeneities may be needed to interrupt transmission and improve treatment outcomes.

## Introduction

Multidrug-resistant tuberculosis (MDR-TB) (i.e., resistance to at least rifampin and isoniazid) poses serious threats to effective tuberculosis (TB) control in many countries. Globally, approximately 4% to 5% of incident TB cases are multidrug resistance (MDR), but this is substantially higher in countries of the former Soviet Union where MDR-TB represents >20% of new TB cases and >50% of previously treated TB [1]. MDR-TB in this region has been attributed to breakdowns in public health infrastructure, transmission of TB in hospitals and prisons, and a deterioration of living conditions coinciding with the dissolution of the Soviet Union in the early 1990s [2]. While the contributions of these factors remain uncertain, there is consensus that the transmission of MDR-TB, as opposed to resistance acquired through inadequate treatment of drug-susceptible TB, is now the predominant cause of incident MDR-TB [3]. This consensus is supported by routine surveillance data that document that the majority of incident MDR-TB episodes are diagnosed among individuals with no prior anti-TB treatment [1]. However, these data alone do not address critical questions about where and between whom MDR-TB is transmitted or reveal the extent to which specific *M. tuberculosis* variants are responsible for MDR-TB transmission.

The increasing availability of next generation sequencing (NGS), coupled with the development of analytic approaches for integrating high-resolution genomic, spatial, and epidemiological data, has transformed our ability to describe transmission of pathogens in populations [4–6]. Previous genomic analyses of TB from the former Soviet Union have described the emergence and evolution of specific *M. tuberculosis* lineages responsible for an outsized proportion of MDR-TB in the region. In general, these studies have been conducted on isolates enriched for drug resistance phenotypes or on samples from larger cohorts [7–9], and this can challenge transmission inference.

We systematically collected and sequenced initial diagnostic isolates from all culture-positive TB cases occurring over 2 years in the Republic of Moldova, a former Soviet country experiencing a severe MDR-TB epidemic. In addition to capturing *M. tuberculosis* isolates from all culture-positive cases, we also collected data on home location and other demographic and epidemiological data, allowing us to study the distribution and dynamics of TB with high resolution across the entire country.

## Methods

### Study setting

Moldova is a small country (approximately 4 million population), which gained independence when the Soviet Union dissolved in 1991. In 2019, the World Health Organization estimated an incidence rate of 80 TB cases (68 to 92) per 100,000 persons. A total of 33% (30% to 35%) of new TB cases and 60% (56% to 64%) of previously treated TB cases were estimated to have MDR-TB [1].

### Study enrollment

TB diagnosis occurs at 46 diagnostic centers located throughout the country. Between January 1, 2018 and December 31, 2019, all nonincarcerated individuals evaluated for pulmonary TB were invited to participate in this study (**Fig A in** S1 Appendix); written consent was provided. This consent allowed us to access routinely collected basic demographic, residential, and epidemiological data and to perform sequencing on their mycobacterial isolates should they have culture-positive TB. This study was approved by the Ethics Committee of Research of the Phthisiopneumology Institute in Moldova and the Yale University Human Investigation Committee (No. 2000023071).

### Data and specimen collection and processing

Demographic data (sex, age, employment, history of incarceration, and education level), residential status (rural or urban residence and home village/locality), and epidemiological data (household contacts and date of diagnosis) were collected from each participant.

Sputum specimens were tested at diagnostic centers by microscopy and Xpert and then sent to 4 in-country laboratories for solid and liquid culture. Positive cultures were sent to the National TB Reference Laboratory in Chisinau for mycobacterial DNA extraction by the cetyl trimethyl ammonium bromide (CTAB) method [10].

### Whole genome sequencing

Genomic DNA was prepared for NGS using the Illumina DNA Prep library preparation kit (S1 Appendix). Raw sequencing files were checked with FastQC [11] and mapped to the H37Rv reference strain (NC_000962.3) using BWA "mem" [12] and sorted with SAMtools v.1.10 [13] (S1 Data). Variant calling was conducted with GATK [14] to identify single nucleotide polymorphisms (SNPs), with low-quality SNPs (Phred score Q <20 and read depth <5) and sites with missing calls in >10% of isolates removed.

Samples with possible polyclonal infections were identified through a previously described method [15] and were not included in the transmission analysis, although we do provide additional details about these polyclonal infections in the Supporting information appendix (S1 Appendix). Heterogenous sites were called as the consensus allele if present in ≥80% of mapped reads; otherwise, they were labeled as ambiguous. SNPs in repetitive regions, PE/PPE genes, and in known resistance-conferring genes were excluded from phylogenetic tree reconstruction. In silico drug resistance prediction was carried out using TB-Profiler v2.8.14 (Tables A–C in S1 Appendix) [16].

### Phylogenetic analysis and transmission cluster identification

A multiple sequence alignment of concatenated SNPs was used to construct a maximum–likelihood (M–L) phylogenetic tree with RAxML [17], using the "GTR-GAMMA" nucleotide substitution model and a Lewis ascertainment bias correction from 500 bootstrap samples.

Putative transmission clusters were identified in the resulting M–L tree using TreeCluster [18], testing 2 distance thresholds of 0.001 and 0.0005 substitutions/site, corresponding to approximate SNP thresholds of 40 and 20, respectively. These thresholds reflect the maximum distance within a cluster; we also estimate the median pairwise distance within a cluster. Timed phylogenetic trees for each large cluster (≥10 cases) identified using the distance threshold of 0.001 substitutions/site were built with BEAST2 v2.6.3. (S1 Appendix) [19]. Briefly, phylogenetic trees were built using a strict molecular clock with a fixed rate of $1.0 \times 10^{-7}$ per site per year and constant population model with a log normal [0,200] prior distribution [20]. Markov chain Monte Carlo chains were run for 250 million iterations, with 10% burn-in to produce maximum clade credibility trees. Finally, past population events in 3 large clades identified in the study population were inferred using the Bayesian Skyline model in BEAST2.

## Inference of person-to-person transmission events

We identified person-to-person transmission events between sampled hosts in large transmission clusters (≥10 cases, TreeCluster distance threshold 0.001 substitutions/site) by reconstructing transmission networks using TransPhylo [21]. This R package uses a Bayesian approach to reconstruct transmission networks from timed phylogenies, including sampled and unsampled hosts, and allows for within-host diversity. We used a "multitree" method that simultaneously infers transmission trees from a selection of input phylogenetic trees while estimating a single value for shared model parameters. This accounts for uncertainty in the phylogenetic tree reconstruction [21]. The procedures for transmission inference within large clusters are detailed in the Supporting information appendix (S1 Appendix).

## Spatial/genetic distance analysis

For each large transmission cluster (≥10 cases), we used a recently developed hierarchical Bayesian regression model to quantify the association between the genetic and spatial distances for unique pairs of cases, adjusting for other pair- and individual-level features and multiple sources of correlation in the data [22]. We then used a Bayesian meta-analysis framework to better understand shared trends and variability in the estimated associations across genetic clusters.

In our main analysis, we modeled the log-scaled patristic distance between each pair of cases within cluster $k$ as a function of geographic distance and other covariates:

$$\ln\left(Y_{kij}\right) = \mathbf{x}_{kij}^{\mathrm{T}} \, \boldsymbol{\beta}_k + \left(\mathbf{z}_{ki} + \mathbf{z}_{kj}\right)^{\mathrm{T}} \boldsymbol{\gamma}_k + \theta_{ki} + \theta_{kj} + \epsilon_{kij}, i < j,$$

where $Y_{kij}$ is the patristic distance between cases $i$ and $j$ within cluster $k$ and $\epsilon_{kij} \sim \mathrm{N}(0, \sigma_{k\epsilon}^2)$ are the independent, Gaussian distributed errors. We defined the expected value as a function of pair- and individual-level information, where $\mathbf{x}_{kij}$ includes covariates based on differences between the pair (i.e., Euclidean distance in kilometers, an indicator for whether the pair is in the same home village/locality, absolute difference between the dates of diagnosis in days, absolute difference between the ages in years) and $z_{ki}$ includes individual-level covariates (i.e., age in years, number of household contacts, sex (male and female), education status (<secondary and ≥secondary), working status (employed and unemployed), residence location type (urban and not urban), and housing status (homeless and not homeless)). The $\theta_{ki}$ are spatially correlated random effect parameters that account for correlation between paired outcomes due to (i) the same individual being represented across multiple paired responses; and (ii) spatial correlation between individuals. Complete details on the statistical model, including prior distributions for the model parameters, are provided in [22].

We fit the regression model separately for each of the transmission clusters with at least 10 cases, using the "Patristic" function in the R package "GenePair" (https://github.com/warrenjl/GenePair). For each individual cluster analysis, we included a predictor if <10% of the values across the pairs were missing and if there were >4 pairs in each of the categorical variable levels, to ensure stable model fitting results. Inference was based on 10,000 samples from the joint posterior distribution after removing the first 10,000 iterations prior to convergence and thinning the remaining 100,000 by a factor of 10 to reduce correlation in the posterior samples.

To better understand shared trends and variability in the estimated associations across genetic clusters, we then used the estimates and uncertainty measures obtained from the first stage analyses within a Bayesian meta-analysis framework. The model for a single association $l$ is given as

$$\hat{\beta}_{kl} | \beta_{kl} \sim \mathrm{N}\left(\beta_{kl}, \hat{\sigma}_{kl}^2\right), k = 1, \ldots, m_l,$$

$$\beta_{kl} \sim \mathrm{N}\left(\mu_{\beta_l}, \sigma_{\beta_l}^2\right),$$

where $\hat{\beta}_{kl}$ is the posterior mean obtained from the regression model fit to cluster $k$ for covariate $l$, $\beta_{kl}$ represents the corresponding true but unobserved value, $\hat{\sigma}_{kl}^2$ is the posterior standard deviation, and $m_l$ is the number of main analyses (out of 35 in total) where covariate $l$ was included. We note that $\gamma_{kl}$ effects are included in this same meta-analysis framework as well but describe the model in terms of $\beta_{kl}$ without loss of generality. We assumed that the true cluster-specific effects arise from a common Gaussian distribution with mean $\mu_{\beta_l}$ and variance $\sigma_{\beta_l}^2$, and estimate these parameters by giving them weakly informative prior distributions such that $\mu_{\beta_l} \sim \mathrm{N}\left(0, 100^2\right)$ and $\sigma_{\beta_l} \sim \mathrm{Uniform}(0, 100)$. By making inference on $\mu_{\beta_l}$ we determined if covariate $l$ had a consistent impact when data were pooled across all clusters and uncertainty in the parameter estimates was correctly quantified. When reporting results from the second stage analysis, we present posterior means and 95% quantile-based credible intervals for $\exp\{\mu_{\beta_{lj}}\}$ (i.e., the pooled effect on the reported as the ratio of expected patristic distances per specified change in covariate value).

As a sensitivity analysis, we repeated these analyses modeling SNP distance (instead of patristic) using a similar negative binomial regression framework (details in S1 Appendix).

## Results

### Study population

We invited all culture-positive TB patients ($N$ = 2770) over the study period to participate; 2,405 consented, and, among them, 2,236 (93%) had available isolates for NGS analysis. These patients lived in 709 named localities within 50 regions (Fig 1). Among enrolled participants with treatment history information ($N$ = 2182, Table 1), 31% had been previously treated for TB, 22% were female, and the median age was 43 years (interquartile range (IQR) 23 to 71). A total of 60% lived in rural regions, and 10% were previously imprisoned.

A total of 779 participants (36%) were infected with genetic variants conferring MDR; 50% (386) of these MDR cases were treatment naive (Table 1). There was substantial geographic variation in distribution of MDR-TB. Transnistria, a small region east of the Dniester River, had localities with the highest proportions of TB cases that were MDR and among the highest incidence rates of MDR-TB in the country (Fig 1, **Fig B in** S1 Appendix).

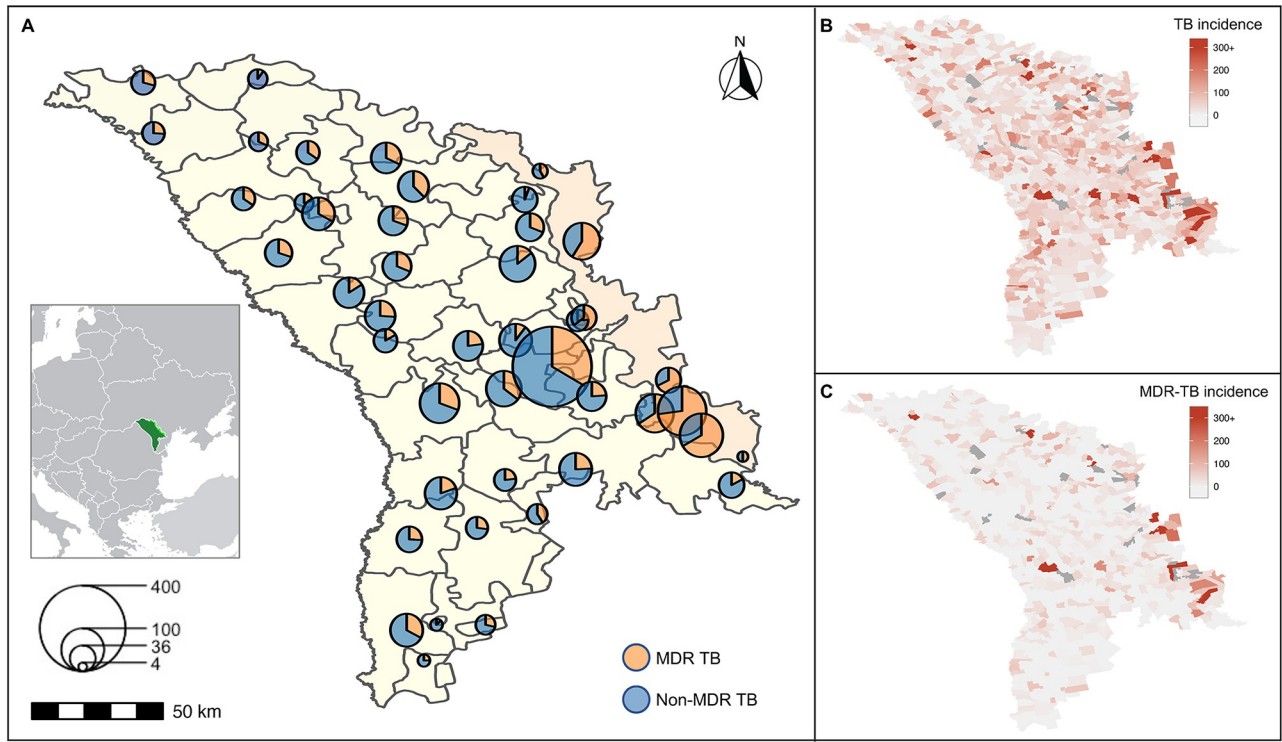

**Fig 1. (A)** Map of culture-confirmed TB patients in Moldova. The center of each circle represents the geometric center of the localities/region (709 named localities within 50 regions) where the case was diagnosed and sampled. The scale indicates the number of culture-confirmed TB patients (*n* = 2,236). The Transnistrian region of Moldova is highlighted. The geographic distribution of the notified incidence of all culture-confirmed **(B)** TB and **(C)** MDR-TB by locality. The colors show the distribution of notified case per population and localities colored dark gray have missing population data. The map data were extracted from the GADM database (www.gadm.org/download_country.html). MDR-TB, multidrug-resistant tuberculosis; TB, tuberculosis.

## Genomic analysis and phylogeny reconstruction

We obtained sequence data from pretreatment specimens of 2,220 participants. Polyclonal infections were identified in 386 participants (17.4%) (**Fig A and C in** S1 Appendix) and removed, resulting in a final dataset of 1,834 *M. tuberculosis* isolates. Among these isolates, 672 (36.6%) were genotypic MDR-TB, including 319 pre-extensive drug resistance (XDR) (17.4%) and 118 XDR (6.4%) TB. Aligning reads against the reference strain revealed 43,284 SNPs that were used to reconstruct a maximum likelihood phylogeny (Fig 2).

A total of 1,014 isolates (55.3%) belonged to Lineage 4 and 804 (43.8%) belonged to Lineage 2/sublineage 2.2.1 (Fig 2A). Mapping revealed distinct geographic patterns for the 3 major MDR-TB clades: clade 1 comprising 243 Ural/lineage 4.2.1 isolates that were widely distributed, and clade 2 and clade 3 containing 102 and 121 Beijing/lineage 2.2.1 strains that were concentrated within Transnistria (Fig 2A). A high proportion of individuals (50.4%) in these 3 large MDR-TB clades had been previously treated for TB.

All Beijing/lineage 2.2.1 strains (802 consensus SNP call, 2 heterogenous SNP call) had a specific nonsynonymous mutation in *esxW* (Thr2Ser), a gene in which mutations were found to be associated with transmission success of Beijing lineages in Vietnam [23]. In contrast, just 3% of non-Beijing strains (32/1,030) harbored this mutation (**Table D in** S1 Appendix). Additionally, 2 nonsynonymous variants in *esxW* were found in low frequencies in non-Beijing strains, 6 samples with a nonsense mutation at codon 172, and 17 samples with a Thr173Ser mutation.

**Table 1. Demographic and clinical characteristics of study participants.**

| Characteristics | All participants (*N* = 2,182*) | New cases (*N* = 1,514) | Previously treated cases (*N* = 668) |
|---|---|---|---|
| *Demographic* | | | |
| Female, no. (%) | 482 (22.1) | 350 (23.1) | 132 (19.8) |
| Age, median (year, IQR) | 43 (23 to 71) | 42 (27 to 65) | 43 (22 to 68) |
| Homeless | 238 (10.9) | 144 (9.5) | 94 (14.1) |
| Rural residence | 1,323 (60.6) | 991 (65.5) | 332 (49.7) |
| Unemployed | 1,437 (65.9) | 995 (65.7) | 442 (66.2) |
| Previously prisoner# | 209 (9.6) | 90 (5.9) | 119 (17.8) |
| Education level | | | |
| Primary | 770 (35.3) | 507 (33.5) | 263 (39.4) |
| Secondary | 1,288 (59.0) | 912 (60.2) | 376 (56.3) |
| No. of household contacts (mean, SD) | 2.30 (2.33) | 2.47 (2.40) | 1.91 (2.11) |
| *Clinical* | | | |
| Smear positive | 975 (44.7) | 697 (46.0) | 278 (41.6) |
| Drug resistance profiles$ | | | |
| Pan-susceptible | 1,152 (52.8) | 939 (62.0) | 213 (31.9) |
| MDR | 779 (35.7) | 386 (25.5) | 393 (58.8) |

* A total of 54 participants did not report the information of TB treatment history.

# A total of 108 participants did not report information of incarceration.

$ The drug-resistant profiles in this table were determined by the whole genome sequencing detection of the drug-resistant–related mutations.

IQR, interquartile range; MDR, multidrug resistance; SD, standard deviation; TB, tuberculosis.

## Prevalence of drug resistance genotypes

The 3 large clades were comprised almost entirely of MDR isolates (96%, 449 of 466) (Fig 2B); resistance-conferring mutations for isoniazid and rifampin were similar and found in the *katG* 315 codon and in the 81-bp rifampicin resistance determination region (RRDR). However, each of these 3 clades had additional distinctive drug resistance mutations: the isolates in Ural strain/lineage 4 clade 1 harbored an *eis* promoter (−12 C>T) mutation conferring kanamycin resistance, one Beijing strain/lineage 2 clade had an *ethA* (110–110 del), associated with ethionamide resistance, while the other had *thyX* (−16 C>T) and *thyA* (Arg222Gly) mutations, associated with resistance to p-aminosalicylic acid. We also identified clusters of isolates harboring additional drug-resistant mutations associated with drugs in newly recommended MDR treatment regimens including lineozid (*n* = 14), bedaquiline (*n* = 1), and delamanid (*n* = 9). We also reported DR mutations among the 386 mixed samples (**Table A** and **B in** S1 Appendix).

## Transmission of drug-resistant *M. tuberculosis*

Of the 1,834 *M. tuberculosis* isolates, 1,551 (85.6%) formed clusters ranging in size from 2 to 105, and 1,000 (54.5%) belonged to 35 large clusters with at least 10 participants at the clustering threshold of 0.001 substitutions/site. The median SNP distance across all transmission clusters was 14 SNPs (IQR 10 to 18 SNPs), with the median within-cluster SNP distance ranging from 0 to 26 SNPs (**Fig D-a in** S1 Appendix). Meanwhile, the median SNP distance in a cluster defined using the threshold of 0.0005 substitutions/site was 9 SNPs (IQR 7 to 12 SNPs) (**Fig D-b in** S1 Appendix).

Of 672 MDR-TB isolates included in the final analysis, 619 (92.1%) were part of a cluster, and 454 (67.6%) belonged to one of the 35 large transmission clusters. Individuals with

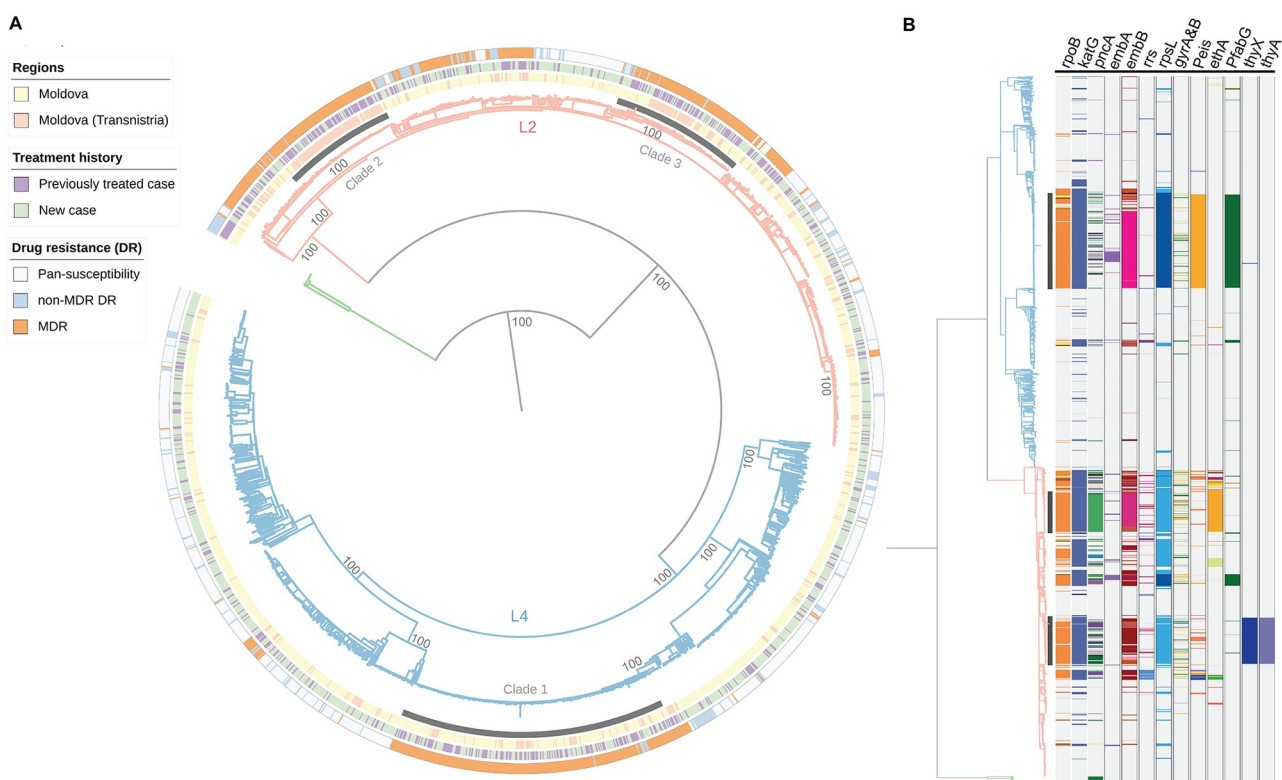

**Fig 2.** (**A**) M–L phylogeny of 1,834 Moldova *M. tuberculosis* isolates based on 43,284 variable sites. The outer bands represent the in silico drug-resistant profiles, treatment history of participant and the region where the isolates were sampled from. The tree is rooted to *Mycobacterium bovis* (branch in green). L2 denotes lineage 2 (light orange) and L4 lineage 4 (light blue). Three major clades from the Ural/ lineage 4.2.1 (clade 1) and Beijing/lineage2.2.1 (clades 2 to 3) are shaded. The main nodes of the tree have 100% bootstrap support. (**B**) Phylogenetic distribution of resistance-related genotypes. The columns depict loci associated with drug resistance. "P" followed by a subscription of gene name indicates the promotor region. Colored bands of each column represent different polymorphisms. DR, drug resistance; MDR, multidrug resistance; MDR-TB, multidrug-resistant tuberculosis; M–L, maximum–likelihood.

MDR-TB were more likely to be in large clusters than individuals with pan-susceptible disease (odds ratio (OR) 3.39, *P*-value < 0.001, **Table E in** S1 Appendix). Eight of the 14 MDR plus linezolid-resistant isolates were members of large clusters (Cluster 1, 2, and 21, **Fig E in** S1 Appendix). Among the 9 MDR isolates with delamanid resistance, 7 had the same delamanid-associated resistance mutation, forming a single subcluster (Cluster 19, **Fig E in** S1 Appendix) with a median pairwise SNP distance of <5 SNPs, suggesting recent transmission of this highly resistant *M. tuberculosis* strain in Moldova.

Closer inspection of the 35 large transmission clusters revealed distinct demographic and epidemiological differences between clusters. The largest transmission cluster (Cluster 1) included 105 participants with the sublineage 4.2.1/Ural Clade 1 stain residing throughout the entire country (Fig 3A and 3D). In contrast, the next largest cluster (Cluster 2) included 102 participants with the sublineage 2.2.1/Beijing Clade 2 stain living predominately in Transnistria (Fig 3B and 3E). A total of 16 of the 35 large clusters were comprised almost entirely of MDR-TB (Fig 3, **Fig E in** S1 Appendix). Notably, there were cluster-specific demographic differences observed across transmission clusters, with the largest 2 groups comprising a high proportion of previous prisoners and reporting unsatisfactory living conditions (**Fig F in** S1 Appendix). **Table E in** S1 Appendix details the association of

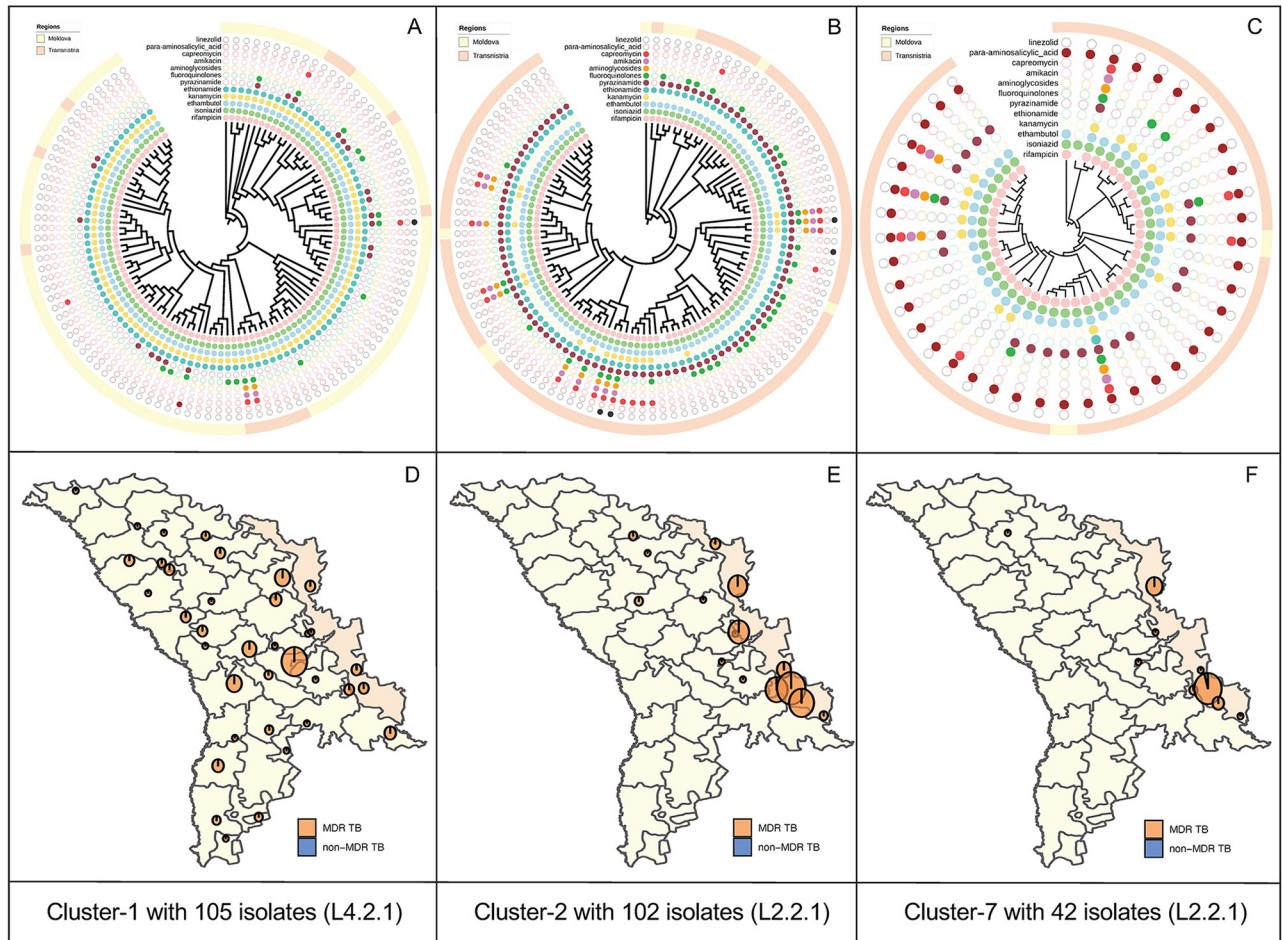

**Fig 3. (A–C)** Tree visualizations for 3 large putative transmission clusters ($N \geq 10$ isolates), each showing the location of cases in either the Moldova or Transnistria regions along with resistance/susceptibility to 12 anti-TB drugs, as identified by in silico prediction. **(D, E)** Spatial distribution of 3 largest clusters (Cluster 1, 2, and 7) in the Ural/Lineage 4.2.1 and Beijing/lineage 2.2.1 clades. The map data were extracted from the GADM database ([www.gadm.org/download_country.html](www.gadm.org/download_country.html)). MDR-TB, multidrug-resistant tuberculosis; TB, tuberculosis.

covariates and membership in large clusters, along with a sensitivity analysis defining clusters using a stricter threshold of 0.0005 substitutions/site that showed broadly the same significant associations.

Reconstructing transmission networks in the 35 broad clusters using the multitree Trans-Phylo approach, we inferred 194 person–person transmission events. The relatively short study period allows for limited opportunities to capture transmission chains and pairs, and, accordingly, a minority of clustered isolates were predicted to be involved in transmission events in at least half the posterior transmission trees (338/1,000, 33.8%). Nonetheless, the identification of these transmission events support evidence of recent, local transmission between sampled individuals in the region. We found no significant factors that were associated with inclusion in these person-to-person transmission events compared to other clustered person-to-person individuals, although there was some evidence for an increased likelihood of transmission linkage between hosts in the Transnistria region compared to the rest of Moldova (OR 1.42, $P = 0.02$).

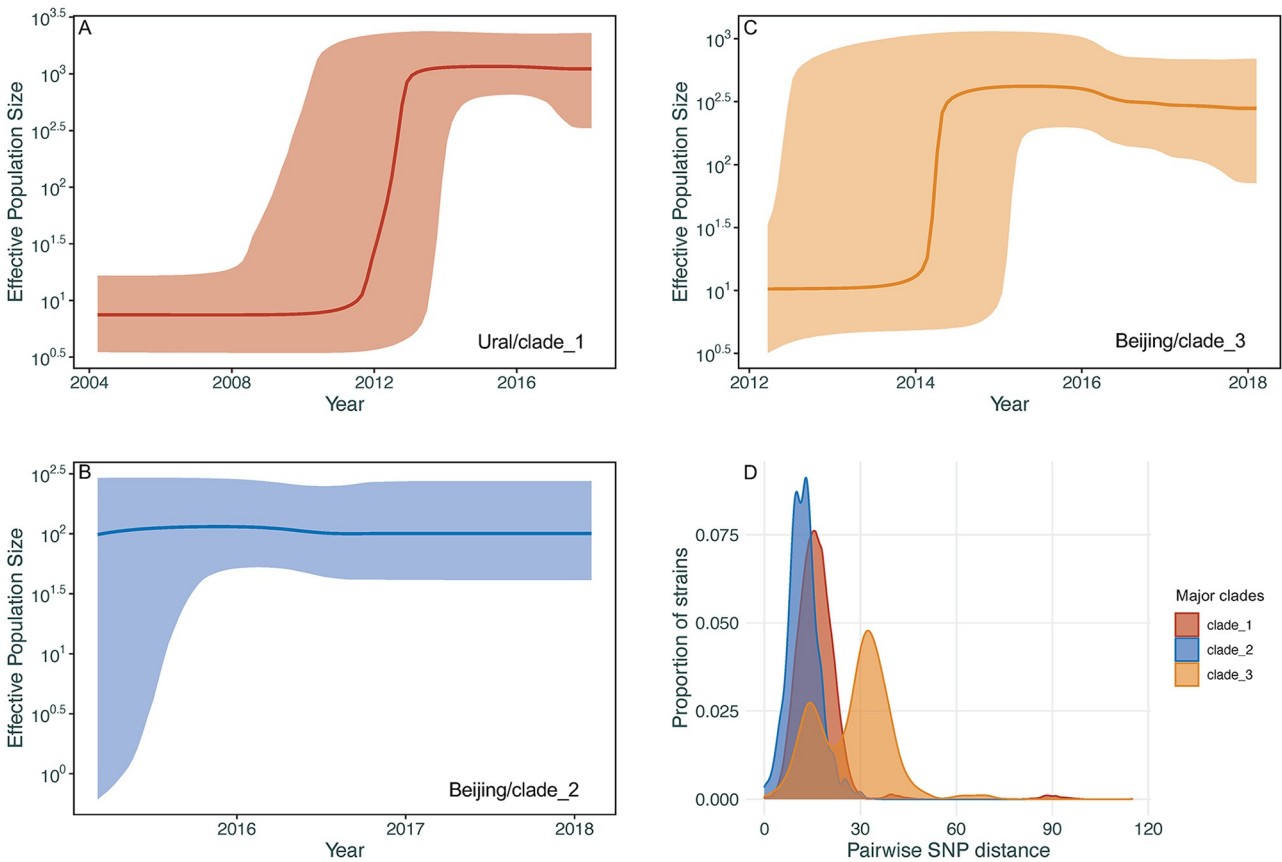

**Fig 4. (A–C)** Coalescent Bayesian Skyline plots of the 3 large clades among Ural/lineage 4.2.1 and Beijing/lineage 2.2.1 with specific resistant mutations (detailed in Fig 2B) using an uncorrelated log normal relaxed clock model. The 2 blue lines are the upper and lower bounds of the 95% HPD interval. The x-axis is the time in years and the y-axis is on a log scale. **(D)** Density distribution of within-clade pairwise SNPs distance of clades 1 to 3. HPD, highest posterior density; SNP, single nucleotide polymorphism.

## Bayesian Skyline analysis of large MDR-TB clades

To gain further insight into the population dynamics of MDR-TB in Moldova, we reconstructed the scaled effective population size for the 3 large MDR-TB clades (Fig 2) and estimated the time to the most recent common ancestor (TMRCA) (**Table F in** S1 Appendix).

We estimated the TMRCA of the Ural/4.2.1 (clade 1) to be around 1984, although with a relatively broad posterior density interval (95% highest posterior density interval (HPDI)) of 1961 to 2003 owing to the limited temporal range of the data. The 2 Beijing/L2.2.1 clades (clades 2 and 3) are estimated to have a TMRCA of 2013 (95% HPDI: 2010 to 2015) and approximately 2006 (95% HPDI: 1999 to 2012), respectively (**Table G in** S1 Appendix), implying more recent introduction of these clades to the region. Fig 4 shows the estimated *M. tuberculosis* effective population size for the 3 major clades over time. Our analysis estimated substantial growth of the Ural/clade 1 between 2012 and 2013 and of the Beijing/clade 3 in late 2013 to mid-2014. For the Beijing/clade 2, the effective population size has remained relatively constant, although the estimated date of origin falls within the time period when growth occurred in other clades. These results indicate a period of population expansion of MDR-TB in Moldova between 2012 and 2014. A sensitivity analysis using alternative clock models and rate estimates (**Table G in** S1 Appendix) showed similar estimates for the TMRCA and effective population sizes for each clade (**Fig G in** S1 Appendix).

**Table 2. Pooled Bayesian meta-analysis inference for each exponentiated effect (i.e., RR interpretation).**

| Effect | Estimate | 95% credible interval |
|---|---|---|
| Distance between localities (50 km) | 1.06 | (1.03, 1.08) |
| Same locality (yes versus no) | 0.53 | (0.40, 0.68) |
| Date of diagnosis distance (1/2 year) | 1.03 | (1.01, 1.07) |
| Age difference (10 years) | 1.00 | (1.00, 1.01) |
| Age (10 years) | 0.99 | (0.97, 1.01) |
| Household contacts (1 person) | 0.99 | (0.97, 1.01) |
| Sex | | |
| Mixed pair versus both female | 0.96 | (0.91, 1.02) |
| Both male versus both female | 0.91 | (0.82, 1.01) |
| Residence location | | |
| Mixed pair versus both not urban | 1.02 | (0.97, 1.09) |
| Both urban versus both not urban | 1.06 | (0.94, 1.21) |
| Housing | | |
| Mixed pair versus both not homeless | 1.02 | (0.87, 1.20) |
| Both homeless versus both not homeless | 1.11 | (0.83, 1.48) |
| Working status | | |
| Mixed pair versus both unemployed | 1.03 | (0.96, 1.12) |
| Both employed versus both unemployed | 1.13 | (0.96, 1.33) |
| Education | | |
| Mixed pair versus both ≥ secondary | 0.98 | (0.94, 1.02) |
| Both < secondary versus both ≥ secondary | 0.95 | (0.87, 1.03) |

Posterior means and 95% quantile-based credible intervals are presented.

RR, risk ratio.

## Spatial/genetic distance analysis

Table 2 shows the pooled risk ratio (RR) inference for pair- and individual-covariates from the Bayesian meta-analysis of genetic and spatial distances. Two cases in the same locality had a 47% lower expected patristic distance compared to cases in different localities (RR: 0.53; 95% CI: 0.40, 0.68). For cases in different localities, as the distance between the localities increases by 50 kilometers, the patristic distance between the pair increased by 6% (RR: 1.06 (1.03, 1.08)). For every half-year increase in the separation between dates of diagnosis for a pair, the patristic distance increased by 3% (RR: 1.03; (1.01, 1.07)). A sensitivity analyses using SNP distances yielded similar results (**Table H in** S1 Appendix).

## Discussion

We describe the recent circulation of 3 distinct clades of *M. tuberculosis* (1 of Ural lineage and 2 of Beijing lineage) responsible for the vast majority of MDR-TB in Moldova. While these clades share similar isoniazid- and rifampin-conferring mutations, there are additional clade-specific mutations conferring resistance to important second-line TB antibiotics critical for MDR treatment success.

Broad transmission networks based on genomic similarity showed that >85% of all culture-positive TB cases in Moldova could be mapped to putative transmission clusters and that the majority (>54%) of these cases were found in 35 large transmission clusters. The role of recent transmission was even more pronounced for MDR-TB cases, among which >92% were found within putative transmission clusters (and >67% found within the 35 large transmission

clusters). Individuals with MDR-TB had over 3-fold higher odds of being in a large transmission cluster compared with individuals with pan-susceptible TB. Other notable covariates associated with increased odds of being in a large transmission cluster included urban residence, previous incarceration, and a history of previous treatment for TB. We found that pairs with closer times of diagnosis and living within the same locality had the greatest genomic similarity and that for pairs in different localities, closer spatial proximity was associated with greater genomic similarity.

Previous analyses of surveillance data have revealed striking spatial heterogeneity of MDR-TB in Moldova with MDR-TB incidence differing by more than an order of magnitude for different localities [24], but the mechanisms driving this variation have not been described. Our analysis reveals that this heterogeneity is associated with the multiple overlapping epidemics of transmitted MDR-TB, some of which are due to clades that have extended across the entire country, while others are thus far confined to specific subregions. Most notably, the 2 largest transmission clusters of the Beijing lineage are found almost exclusively in Transnistria, where, in some localities, MDR-TB incidence rates exceed 200 cases per 100,000 persons/year. Our finding that nearly all Beijing lineage strains in Moldova have *esxW* mutations corroborates recent work that suggests that these variants may be under positive selection [23].

A recently reported genomic study conducted among patients diagnosed in 2013 and 2014 at a single municipal hospital in Chisinau described the local concentration of MDR-enriched lineage 4.2.1 (Ural) isolates [25]. In the current study, conducted approximately 6 years later and inclusive of the entire country, we found that MDR isolates within this lineage are present throughout Moldova and are commonly within transmission clusters, although this has thus far only been reported sporadically outside Moldova [26]. Prior work had found this lineage to be responsible for MDR-TB due to reinfection in nosocomial settings [27]; it is now apparent that these MDR strains are transmitted frequently in community settings. Regional reviews have suggested an important role of Beijing and Ural lineages in current TB epidemics [28]; our current work confirms and builds upon these insights, revealing in high resolution the overlapping dynamics of these 2 lineages in Moldova.

A major strength of our study was that we were able to include all culture-positive isolates across the country, minimizing challenges to transmission inference due to sampling biases. However, because we only could collect samples for 2 years—a short duration compared with the natural history of TB—our ability to track chains of transmission and to predict who infected whom was limited. We cannot rule out bias caused by individuals with TB that were never diagnosed or because some TB cases were not culture positive [29]. Additionally, polyclonal samples were removed from this analysis due to difficulties in producing well-resolved phylogenies. We do note that we found evidence for homogeneous and heterogenous drug resistance mutations in these sequences at a similar proportion to the remaining study population (**Table B in** S1 Appendix). Further methods development and analysis are required to understand the potential role of polyclonal TB infection in transmission within Moldova.

There are urgent clinical and public health implications of these findings. While the crisis of transmitted MDR-TB was already apparent in this region, these data reveal that there are several cocirculating highly drug-resistant TB clades that differ in terms of drug resistance profiles, geographic distribution, and epidemic trajectory. These results suggest the urgency of interrupting MDR-TB transmission in Moldova, especially within specific geographic foci in the capital city of Chisinau and in the region of Transnistria. While the role of genomic surveillance for informing TB interventions in high-burden settings remains incompletely explored, this study provides an important example of how such information may be used to understand the complex epidemiology of MDR-TB in a high incidence country. We must next investigate

whether this improved understanding of local transmission can inform the design of more effective and efficient interventions, a question which remains unanswered at this time.

## Supporting information

**S1 STROBE Checklist. STROBE Statement—Checklist of items that should be included in reports of observational studies.** STROBE, STrengthening the Reporting of OBservational studies in Epidemiology.
(DOCX)

**S1 Data. Additional demographic and epidemiological data used in the analysis.**
(CSV)

**S1 Appendix. Table A:** A summary of the lineages found in mixed M. tuberculosis samples from Moldova, as designated by TB-Profiler. **Table B:** A summary of the homogeny in drug resistance mutations present in mixed M. tuberculosis samples from Moldova. **Table C:** In silico drug resistance prediction using TBprofiler and genTB tools. **Table D:** Allele counts for 9 SNP variants identified in the esxW gene within the study population, showing counts within samples classified as either Beijing strains (all lineage 2.2.1) or as any other lineage. **Table E:** Demographic associations in cases belonging to large transmission clusters (10 cases), identified with patristic distance thresholds of 0.001 and 0.0005. Cases in small clusters (2 to 9 cases) are not included. ORs are calculated using logistic regression and P values by Wald chi-squared test, adjusted for age and sex. **Table F:** Results of the Coalescent Bayesian Skyline analyses of the 3 large clades with specific resistant mutations using an uncorrelated log normal relaxed clock model. **Table G:** Complete Coalescent Bayesian Skyline results of the sensitivity analysis using 3 different clock model settings (strict, log normal relaxed, and exponential relaxed) and 3 clock rate estimates of the 3 large clades with specific resistant mutations. The clock rate used log normal distribution. **Table H:** Pooled Bayesian meta-analysis inference for each exponentiated effect (i.e., ratio of expected SNP distances per specified change in covariate value). Posterior means and 95% quantile-based credible intervals are presented. **Fig A:** The study flow diagram. **Fig B:** Distribution of the proportion of MDR-TB by the regions where they were diagnosed. (a) Regions sorted by the proportion of MDR-TB and (b) the total numbers of MDR-TB isolates from high to low. **Fig C:** (a) A scatterplot showing the pairwise SNP distance (max. 50 SNP differences) plotted against the patristic distance on an M–L phylogeny produced with RAxML between all 2,236 Moldovan isolates with whole genome sequence data. (b) A scatterplot showing the pairwise SNP distance (max. 50 SNP differences) plotted against the patristic distance on an M–L phylogeny produced with RAxML between 1,834 nonmixed Moldovan isolates with whole genome sequence data. **Fig D:** (a) The pairwise SNP distance in 35 large transmission clusters with at least 10 participants involved with the threshold of 0.001. The box plot shows the IQR and median SNP distance of each cluster. (b) The pairwise SNP distance in 26 large transmission clusters with at least 10 participants involved with the threshold of 0.0005. The box plot shows the IQR and median SNP distance of each cluster. **Fig E:** Tree visualizations for remaining 32 transmission clusters (N ≥ 10 isolates), each showing the location of cases in either the Moldova or Transnistria regions along with resistance/susceptibility to anti-TB drugs, as identified by in silico prediction. **Fig F:** Tree visualizations for the 35 transmission clusters (N ≥ 10 isolates), each showing the location of cases in either the Moldova and Transnistria regions along with selected covariates, namely, urban residence, homeless, unsatisfactory living conditions, and former prisoner. **Fig G:** Coalescent Bayesian Skyline plots of the sensitivity analysis using 3 different clock model settings (strict, log normal relaxed, and exponential relaxed) and 3 clock rate estimates of the 3 large

clades with specific resistant mutations. IQR, interquartile range; MDR-TB, multidrug-resistant tuberculosis; M–L, maximum–likelihood; OR, odds ratio; SNP, single nucleotide polymorphism.
(PDF)

## Acknowledgments

We thank the clinical and laboratory staff of Phthisiopneumology Institute from Chisinau and Regional Reference Laboratories from Balti, Vorniceni and Bender from Moldova for invaluable help and for their assistance in collecting and testing patient specimens.

### Disclaimers

The contents are the responsibility of the authors and Subgrantee and do not necessarily reflect the views of USAID or the United States Government.

## Author Contributions

**Conceptualization:** Chongguang Yang, Benjamin Sobkowiak, Alexandru Codreanu, Valeriu Crudu, Caroline Colijn, Ted Cohen.

**Data curation:** Chongguang Yang, Alexandru Codreanu, Nelly Ciobanu, Melanie H. Chitwood, Marcus Russi, Joshua Havumaki, Patrick Cudahy, Heather Fosburgh, Christopher J. Allender, Heather Centner, David M. Engelthaler, Nicolas A. Menzies, Joshua L. Warren, Valeriu Crudu, Ted Cohen.

**Formal analysis:** Chongguang Yang, Benjamin Sobkowiak, Vijay Naidu, Joshua L. Warren, Caroline Colijn, Ted Cohen.

**Funding acquisition:** Chongguang Yang, Heather Fosburgh, Nicolas A. Menzies, Valeriu Crudu, Caroline Colijn, Ted Cohen.

**Investigation:** Chongguang Yang, Benjamin Sobkowiak, Vijay Naidu, Alexandru Codreanu, Melanie H. Chitwood, Sofia Alexandru, Stela Bivol, Marcus Russi, Joshua Havumaki, Patrick Cudahy, Christopher J. Allender, Heather Centner, David M. Engelthaler, Nicolas A. Menzies, Valeriu Crudu, Caroline Colijn, Ted Cohen.

**Methodology:** Chongguang Yang, Benjamin Sobkowiak, Vijay Naidu, Kenneth S. Gunasekera, Joshua L. Warren, Ted Cohen.

**Project administration:** Alexandru Codreanu, Nelly Ciobanu, Sofia Alexandru, Stela Bivol, Heather Fosburgh, Valeriu Crudu, Caroline Colijn, Ted Cohen.

**Resources:** Alexandru Codreanu, Nelly Ciobanu, Sofia Alexandru, Stela Bivol, Heather Fosburgh, Nicolas A. Menzies, Ted Cohen.

**Software:** Chongguang Yang, Benjamin Sobkowiak, Vijay Naidu, Christopher J. Allender, Heather Centner, David M. Engelthaler, Joshua L. Warren, Caroline Colijn.

**Supervision:** Stela Bivol, Valeriu Crudu, Caroline Colijn, Ted Cohen.

**Validation:** Chongguang Yang, Benjamin Sobkowiak, Vijay Naidu, Kenneth S. Gunasekera, Melanie H. Chitwood, Sofia Alexandru, Christopher J. Allender, Valeriu Crudu, Ted Cohen.

**Visualization:** Chongguang Yang, Benjamin Sobkowiak, Vijay Naidu, Joshua L. Warren, Ted Cohen.

**Writing – original draft:** Chongguang Yang, Benjamin Sobkowiak, Caroline Colijn, Ted Cohen.

**Writing – review & editing:** Chongguang Yang, Benjamin Sobkowiak, Vijay Naidu, Alexandru Codreanu, Nelly Ciobanu, Kenneth S. Gunasekera, Melanie H. Chitwood, Sofia Alexandru, Stela Bivol, Marcus Russi, Joshua Havumaki, Patrick Cudahy, Heather Fosburgh, Heather Centner, David M. Engelthaler, Nicolas A. Menzies, Joshua L. Warren, Valeriu Crudu, Caroline Colijn, Ted Cohen.

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
