## [Editor Report · Decision Letter 0]

21 Jul 2021

Dear Dr Yang, 

Thank you for submitting your manuscript entitled "Phylogeography and transmission of M. tuberculosis in Moldova" for consideration by PLOS Medicine.

Your manuscript has now been evaluated by the PLOS Medicine editorial staff and I am writing to let you know that we would like to send your submission out for external peer review.

Please re-submit your manuscript within two working days, i.e. by Jul 23 2021 11:59PM.

Kind regards,

Beryne Odeny

Associate Editor

PLOS Medicine

---

## [Decision Letter · Decision Letter 1]

7 Oct 2021

Dear Dr. Yang,

Thank you very much for submitting your manuscript "Phylogeography and transmission of M. tuberculosis in Moldova" (PMEDICINE-D-21-03120R1) for consideration at PLOS Medicine. 

[LINK]

In light of these reviews, I am afraid that we will not be able to accept the manuscript for publication in the journal in its current form, but we would like to consider a revised version that addresses the reviewers' and editors' comments. Obviously we cannot make any decision about publication until we have seen the revised manuscript and your response, and we plan to seek re-review by one or more of the reviewers. 

We expect to receive your revised manuscript by Oct 28 2021 11:59PM. Please email us (plosmedicine@plos.org) if you have any questions or concerns.

We look forward to receiving your revised manuscript. 

Sincerely,

Beryne Odeny, 

PLOS Medicine

plosmedicine.org

1) Please revise your title according to PLOS Medicine's style. Your title must be nondeclarative and not a question. It should begin with main concept if possible. Please place the study design in the subtitle (i.e., after a colon). For example, “Phylogeography and transmission of M. tuberculosis in Moldova: A prospective genomic analysis” 

2) The Data Availability Statement (DAS) requires revision. If part of the data is not freely available, please include an appropriate contact (web or email address) for inquiries (this cannot be a study author/ co-author).

3) At this stage, we ask that you write a non-technical Author Summary. The Author Summary should immediately follow the Abstract in your revised manuscript. This text is subject to editorial change and should be distinct from the scientific abstract. The summary should be accessible to a wide audience that includes both scientists and non-scientists. Please see our author guidelines for more information: https://journals.plos.org/plosmedicine/s/revising-your-manuscript#loc-author-summary.

4) Abstract:

a) Please ensure that all numbers presented in the abstract are present and identical to numbers presented in the main manuscript text.

b) In the last sentence of the Abstract Methods and Findings section, please describe the main limitation(s) of the study's methodology.

5) Did your study have a prospective protocol or analysis plan? Please state this (either way) early in the Methods section:

 a) If a prospective analysis plan (from your funding proposal, IRB or other ethics committee submission, study protocol, or other planning document written before analyzing the data) was used in designing the study, please include the relevant prospectively written document with your revised manuscript as a Supporting Information file to be published alongside your study and cite it in the Methods section. A legend for this file should be included at the end of your manuscript. 

6) Please specify whether informed consent was written or oral.

7) Please ensure that the study is reported according to the STROBE guideline for observational studies, and include the completed STROBE checklist as Supporting Information. Please add the following statement, or similar, to the Methods: "This study is reported as per the Strengthening the Reporting of Observational Studies in Epidemiology (STROBE) guideline (S1 Checklist)." The STROBE guideline can be found here: http://www.equator-network.org/reporting-guidelines/strobe/

8) Your study is observational and therefore causality cannot be inferred. Please remove language that implies causality throughout the manuscript, such as “... the result of...” For example, you state in line 396-397 that “Our analysis reveals that this heterogeneity is the result of…” Please refer to associations instead.

9) In the Methods and Results section, please consistently provide 95% CIs and p values for estimates in the main text and tables.

10) Figures 1-3, S4-S5:

a) Please ensure that Figure 1 complies with our figure requirement: http://journals.plos.org/plosmedicine/s/figures. 

b) Please confirm that the appropriate usage rights apply to the use of this Figure 1. Please see our guidelines for map images: https://journals.plos.org/plosmedicine/s/figures#loc-maps

11) Please do not report P<0.01, instead report as P < 0.001.

12) Please remove the “Role of the funding source”, “Declaration of interests”, and “Data sharing” from the main text. This information will be published as metadata based on your responses to the submission form.

13) Please use PloS Medicine style in-text reference call outs noting the square brackets. For example "... countries [1,2]."

Comments from the reviewers:

Reviewer #1: General Comments

Yang and colleagues present the results of a prospective genomic analysis of all TB cases in the Republic of Moldova diagnosed in 2018 and 2019. The authors used next generation sequencing and phylogenetic methods to determine the proportion of cases attributable to multi-drug resistant M. tuberculosis (MDR-TB) variants and characterize their genetic clustering and spatial distribution. Finally, they identified independent individual and geographic predictors of recent transmission of MDR-TB. The authors conclude that the MDR-TB epidemic in Moldova is driven by local transmission and that there is an urgent need to apply comprehensive genomic surveillance to understand transmission and potentially to inform the design of public health interventions to contain the DR-TB epidemic.

Major Comments

This is a rigorously performed study that seeks to answer a provocative, if open-ended question, about what can be learned when demographic, spatiotemporal, and pathogen genomic data are integrated together at population level to provide a broad picture of an MDR-TB epidemic in a high-incidence country.

I have a few brief questions and comments about the methods, presentation, and interpretation, as well as a larger question about the place of this work in the literature and its implications.

1) The authors mention the comprehensive sampling approach as a strength and note that some prior studies did not do this, including a recent study from Moldova. Is it possible to comment on how the inferences changed empirically with comprehensive sampling, and if so how? Are there any alternatives (e.g. stratified random samples) to comprehensive sampling that could improve feasibility without sacrificing rigor?

2) Reduced reagent volumes were one of the techniques the authors used to improve the affordability of comprehensive WGS. Should this not be highlighted as a strength or innovation of the study? Are the other learnings to improve feasibility that might inform replication of this study for research or practice?

3) Can the authors state somewhere why was Moldova chosen as a location for this study? Can the authors comment on the possible generalizability of these findings to other countries with DR-TB epidemics, in the region and possibly beyond? Presumably every country has its unique features, but are there particular characteristics of Moldova (beyond its small size) or of its MDR-TB epidemic that made it more suitable for this project than others?

4) The authors present a detailed analysis of pathogen and demographic factors that may drive transmission, but there is also very limited clinical information about host factors. Could the authors speculate to what extent could HIV, or other host factors that also cluster in individuals, might act as confounders of the observed associations? Should this type of clinical information be captured in a comprehensive surveillance system?

5) Can the authors comment on the potential effects of excluding polyclonal infections? I understand that this is technically necessary, but it would help to hear the implications for the overall analysis. For example, to the extent that mixed infections reflect greater transmission, could the overall estimates of recent and local transmission be underestimates?

6) Some additional explanation to help contextualize these findings would help highlight its importance to general audiences. Specifically, the application of TB molecular epidemiology methods to improve global TB control has been somewhat disappointing. To what extent are the two innovations described here, whole genome sequencing and comprehensive surveillance, well-positioned to overcome some of the limitations of prior approaches, and, if adopted, better inform design of interventions to improve TB control and elimination? What are the next steps, scientifically, to move from the detailed understanding provided in this paper to such interventions?

Minor Comments:

Results

Could the authors report the prevalence of pre-XDR and XDR isolates? Figure 2 suggests that the number of rrs and rlsb mutations may be large but the number of gyrA mutations appears small.

Figures and Tables

Figure 1. Please state the number of participants shown in the figure, as well as the total number of districts and regions. Could you elaborate on "The scheme of color represents the

distribution of notified incidence and localities with missing population data are in gray." Most of the figure(s) are gray - did you really mean missing denominators, or missing cases (as in no cases reported?), which is what the legend implies?

A flow diagram, as specified by STROBE guidelines, is needed to aggregate in one figure all data on patient inclusion in and exclusion from analyses. This could appear in the Supplement but should be referenced in the main text. The inclusion of a high proportion of the eligible population is actually a strength of the study.

Supplement

There appears to be a discrepancy in the number of polyclonal infections reported in the text (n=386) and the number of mixed infections in the supplement (n=403) - I assume that these should be the same.

The legend and other aspects of Table S4 on Page S18 printed partly outside the margins.

Reviewer #2: General comments

The manuscript is clearly described and provide additional insight into the MDR-TB transmitted by heterogeneity is the result of multiple overlapping epidemic in Moldova. 

The subject of the manuscript is of importance as 3 main clades of M.tb transmission have been well addressed in the study,the finding could be useful for public health response to address this issue.

Major comments

1.the SNP threshold is crucial to define the cluster and explore the transmission, given in the high TB epidemic setting, the genetic evolution might be varied under the different selection pressures, so for the genomic similarity, it will be reasonable if the author describe why the SNP thresholds of 40 and 20 were chosen, rather than 12 or 18. As we know, under these thresholds and the cluster rates tend to be decreased accordingly. the author had better make analysis and add relevant content in the paper.

Minor comments

1.Within this manuscript the authors have looked at whole picture of Moldova, in this study, the author have collected all culture-positive isolate in Moldova, however, infection and development of TB should be pointed out as the limitation and bias of the un- recovered strain and missing case. 

2. inference of direct transmission it seems solid analysis, however, in this study, as to chronic infectious disease with relatively unclear exposure history, how about the in-direct transmission.

Reviewer #3: The study is relevant to understanding the distribution and dynamics of TB, with important approaches in phylogenetic analysis and transmission cluster identification as well as in genomic analysis and phylogeny reconstruction. The methodology used is consistent with the objective of the study, which was approved by the Ethics Committee and has a consenting participant enrolment.

The results showed in detail the Map of culture-confirmed TB patients in Moldova, the phylogenetic distribution of resistance-related genotypes, the prevalence of drug resistance genotypes, the transmission of drug-resistant M. tuberculosis, and the estimate of M. tuberculosis effective population size for the three major clades over time, which collaborates to intervene in the dynamics of disease transmission. The tables present relevant data (demographic and clinical characteristics of study participants/pooled bayesian meta-analysis inference for each effect on the relative risk scale) and the figures clearly present the results of the study.

The discussion was carried out with a scientific basis and with a description of the strength and limitation of the study, which corroborates for further studies to be carried out and provides an important example of how such information may be used to understand the complex epidemiology of MDR-TB in a high incidence country.

I congratulate the authors for their initiative in this study and their scientific collaboration in TB control.

Reviewer #4: The authors of "Phylogeography and transmission of M. tuberculosis in Moldova" have done a great job at elucidating the phylogenetic intricacies of TB spread in Moldova, using robust methodologies for phylogeny inference. The biggest strength of this study is that it includes all TB-positive samples from the whole country of Moldova, and hence provides a good overview of the situation concerning tuberculosis in the country. In general, this study is well designed, implemented, and written. However, the huge amount of whole genome sequence data generated by this study could yield more informative results if analyzed a bit further. Kindly find bellow my suggestions.

A- On line 132 the authors state that in silico drug resistance prediction was carried out using TB-Profiler. Other studies exploring the in silico drug resistance genotypes of TB have noted that the use of more than one software to predict these genotypes would yield complementary results. Although I share the methodological viewpoint of the authors that currently TB-Profiler holds the throne for in silico drug resistance genotyping of TB with its huge database and robust methodology, I am curious to know whether the authors tested a few other bioinformatic tools such as Mykrobe or KvarQ. 

Additionally, a study of this scale would benefit from the implementation of a newly developed methodology that reports to have a slightly higher sensitivity than TB-Profiler. Although still in draft form, the tool GenTB reportedly has 3.2% higher sensitivity compared to TB-Profiler in detecting resistance conferring mutations for the nine second-line anti-TB drugs. Seeing as how this could mean that up to 71 samples from this study could have their resistance profiles altered, I suggest that the authors implement the approach outlined in [https://github.com/farhat-lab/gentb-site] and report any changes to their in silico drug resistance prediction results.

B- On line 127, and in more detail in the supplementary data, the authors state that samples exhibiting heterogeneity were discarded from the analysis. Although their aim to minimise the discordance between genomic variation phylogenetic relatedness clarifies as to why 18% of their samples were removed from the phylogenetic analyses, nevertheless, I believe that there is potential in analysing these 403 heterogeneous samples. 

Other publications have shown that polyclonal TB infections are of clinical interest as highlighted in (https://dx.doi.org/10.3201%2Feid2311.170077) this paper also senior authored by Dr. Cohen. Therefore, I suggest that these 403 samples be checked for the presence of mutations encoding drug resistance, and reported in a table format without a phylogeny simply for the added value of this analysis to report the MDR status of polyclonal infections in Moldova. Consequently, there should be a brief discussion on the importance of polyclonal TB infections and their clinical implications 

https://www.nature.com/articles/srep41410

https://journals.plos.org/plosone/article?id=10.1371/journal.pone.0237345

https://www.sciencedirect.com/science/article/pii/S1472979217302251

https://www.sciencedirect.com/science/article/pii/S1472979221000706

I hope that these suggestions aid in improving your already well-drafted manuscript

Balig Panossian

Reviewer #5: Alex McConnachie, Statistical Review

The paper by Yang and colleagues looks at the genetics of tuberculosis cases in Moldova in 2018/19. Much of the paper is beyond my area of expertise, but I can comment on the spatial/genetic distance analysis described from line 162 onwards.

This analysis takes each pair of cases within a cluster as the unit of analysis, and treats the patristic distance between them as the outcome. Again, this is not my field, but I believe this is a measure of the "genetic distance" between two cases. The predictors are measures of within-pair differences, in space, time, or demographics, and some individual demographic measures.

The model assumes that the log-transformed patristic distance is a linear function of the predictor variables, with Normally-distributed, independent errors. The authors do not give any indication of whether these assumptions about the error part of the model are reasonable. I would be particularly concerned about the independence assumption, since the outcomes are derived from pairs of cases. I would expect this to lead to some in-built correlation between residuals. In general, failure to account for correlations between residuals will tend to result in overly-precise estimates of associations.

The results of these analyses are combined across clusters using a Bayesian meta-analysis, which is good. However, the associations are reported as relative risks, which seems wrong, since the outcome is continuous. I am never quite sure of the best term to use, but the exponentiated regression coefficients could perhaps be described as ratios of geometric means associated with a given change in a predictor, but if the authors can think of a better descriptor, that would be fine.

Similar comments could be made about the Poisson version of this analysis reported in the appendix. Are the errors really independent? Is the Poisson distribution assumption (mean = variance) reasonable? Is "relative risk" a sensible term to describe the estimated associations?

One final comment - logistic regression is used to look at factors associated with a case being in a "large cluster", but this is not described in the methods section.

[LINK]

---

## [Decision Letter · Decision Letter 2]

12 Jan 2022

Dear Dr. Yang,

Thank you very much for re-submitting your manuscript "Phylogeography and transmission of M. tuberculosis in Moldova: A prospective genomic analysis" (PMEDICINE-D-21-03120R2) for review by PLOS Medicine.

I have discussed the paper with my colleagues and the academic editor and it was also seen again by four reviewers. I am pleased to say that provided the remaining editorial and production issues are dealt with we are planning to accept the paper for publication in the journal.

[LINK]

We look forward to receiving the revised manuscript by Jan 12 2022 11:59PM.   

Sincerely,

Beryne Odeny, 

PLOS Medicine

plosmedicine.org

Requests from Editors:

1. Please provide a web link to access GenBank data (PRJNA736718)

2. Author summary – Regarding the content under “What did the researchers do and find?” please trim it down to 3 or 4 bullet points. 

3. Your study is observational and therefore causality cannot be inferred. Please remove language that implies causality throughout the manuscript abstract, summary, main text and conclusions, such as, “is fueled by …” Please refer to associations, e.g., “…is associated with …” or similar

4. The terms gender and sex are not interchangeable (as discussed in http://www.who.int/gender/whatisgender/en/ ); please use the appropriate term

5. Please temper claims of primacy of results (e.g., the first study to…) by stating, "to our knowledge" or something similar. 

6. Thank you for providing your STROBE checklist. Please replace the page numbers with paragraph numbers per section (e.g. "Methods, paragraph 1"), since the page numbers of the final published paper may be different from the page numbers in the current manuscript.

7. Please provide the meaning of abbreviations used in tables and figures e.g. yr, IQR, SD, MDR. Please provide this in the footnotes. 

Comments from Reviewers:

Reviewer #1: The Authors have addressed my previous questions and comments in their responses. I offer my congratulations on their work.

Reviewer #2: The author have asked and modified manuscript according to the comments, i feel comfortable about this version.

Reviewer #4: The authors have worked diligently to address the suggestions put forth by the other reviewers and myself. I have no further comments on this polished version of the manuscript

Reviewer #5: Alex McConnachie, Statistical Review

I thank the authors for their responses to my original points, which were more than satisfactory. I am sorry that they had to do all that extra work to fit a model that made little difference to the final results!

I noticed a possible typo. On line 219, just before the regression equation, the sentence ends "...and other covariates such as" which sounds as if something is missing.

I have no further comments.

[LINK]

---

## [Editor Report · Decision Letter 3]

25 Jan 2022

Dear Dr. Yang,

Thank you very much for re-submitting your manuscript "Phylogeography and transmission of M. tuberculosis in Moldova: A prospective genomic analysis" (PMEDICINE-D-21-03120R3) for review by PLOS Medicine.

I am pleased to say that provided the remaining editorial and production issues are dealt with we are planning to accept the paper for publication in the journal.

[LINK]

We look forward to receiving the revised manuscript by Feb 01 2022 11:59PM.   

Sincerely,

Beryne Odeny, 

PLOS Medicine

plosmedicine.org

Requests from Editors:

1. Please replace the term "gender" with "sex." PLOS Medicine style is to use biological "sex" rather than "gender", where appropriate. Apologies for the confusion caused with the terms used.

[LINK]

---

## [Editor Report · Decision Letter 4]

31 Jan 2022

Dear Dr Yang, 

On behalf of my colleagues and the Academic Editor, Dr. Claudia M. Denkinger, I am pleased to inform you that we have agreed to publish your manuscript "Phylogeography and transmission of M. tuberculosis in Moldova: A prospective genomic analysis" (PMEDICINE-D-21-03120R4) in PLOS Medicine.

PRESS

Sincerely, 

Beryne Odeny 

PLOS Medicine